# LDHA Suppression Altering Metabolism Inhibits Tumor Progress by an Organic Arsenical

**DOI:** 10.3390/ijms20246239

**Published:** 2019-12-11

**Authors:** Yu-Jiao Liu, Xiao-Yang Fan, An-Dong Wang, Yin-Zheng Xia, Wen-Rong Fu, Jun-Yi Liu, Feng-Lei Jiang, Yi Liu

**Affiliations:** 1Key Laboratory of Analytical Chemistry for Biology and Medicine (Ministry of Education), Sauvage Center for Molecular Sciences, College of Chemistry and Molecular Sciences, Wuhan University, Wuhan 430072, China; yujiaoliu@whu.edu.cn (Y.-J.L.); xiaoyangfan88@gmail.com (X.-Y.F.); 2015301040084@whu.edu.cn (A.-D.W.); 2015301040203@whu.edu.cn (Y.-Z.X.); fwr1278394447@163.com (W.-R.F.); fljiang@whu.edu.cn (F.-L.J.); 2Department of Biology, University of Maryland, College Park, MD 20742, USA; julieliu02@gmail.com; 3Guangxi Key Laboratory of Natural Polymer Chemistry, College of Chemistry and Materials Science, Nanning Normal University, Nanning 530001, China; 4Hubei Province Key Laboratory for Coal Conversion and New Carbon Materials, School of Chemistry and Chemical Engineering, Wuhan University of Science and Technology, Wuhan 430081, China

**Keywords:** organic arsenicals, lactate dehydrogenase A, metabolism, ROS, mitochondria, apoptosis

## Abstract

Based on the potential therapeutic value in targeting metabolism for the treatment of cancer, an organic arsenical PDT-BIPA was fabricated, which exerted selective anti-cancer activity in vitro and in vivo via targeting lactate dehydrogenase A (LDHA) to remodel the metabolic pathway. In details, the precursor PDT-BIPA directly inhibited the function of LDHA and converted the glycolysis to oxidative phosphorylation causing ROS burst and mitochondrial dysfunction. PDT-BIPA also altered several gene expression, such as HIF-1α and C-myc, to support the metabolic remodeling. All these changes lead to caspase family-dependent cell apoptosis in vivo and in vitro without obvious side effect. Our results provided this organic arsenical precursor as a promising anticancer candidate and suggested metabolism as a target for cancer therapies.

## 1. Introduction

Tumors always show glycolytic “addiction” for enhanced glucose uptake and reduced oxidative phosphorylation to produce ATP. That is, clinical therapies could benefit from altering tumor metabolism from glycolysis to oxidative phosphorylation or inhibiting it. LDHA (lactate dehydrogenase A) is frequently upregulated in clinical tumors to support the high glycolytic activity, and high expression has been thought to be responsible for many cancers, such as glioblastoma, pancreatic, and some kidney cancers. LDH inhibitors could fulfill these unmet therapeutic needs.

LDH has five isoforms LDH1 (B4), LDH2 (B3A1), LDH3 (B2A2), LDH4 (B1A3), and LDH5 (A4) encoded by the *LDHA* and *LDHB* genes. LDH5 (LDHA) is a key glycolytic enzyme that catalyzes the formation of lactate from pyruvate, while LDH1 (LDHB) catalyzes the back formation [1]. 30% ATP production comes from glucose (glycolysis and oxidation) and 10% from glutamine. It was considered that lactate contributed to the other oxidative fuel source [2].

In recent years, for the development of novel anti-cancer agents, therapeutic strategies investigation has been conducted through targeting substantially altered cellular metabolism. Cancer metabolic rewiring facilitates tumor development and/or progression by affecting epigenetics and cell fate decisions through the regulation of metabolic enzymes [3]. Researchers showed immense interest of getting agents which could selectively eradicate cancer cells by altering metabolism [4]. However, few specific LDHA inhibitors complied with the envisaging results in vivo. Oxamate, a pyruvate analog that inhibits LDH activity by blocking the pyruvate binding site, is a weak inhibitor (*IC*_50_ ~ 800 μM) and lacks selectivity. GNE-140 is a selective LDHA inhibitor of nano-molar potency but if it is removed from the medium, cells can proliferate even after 2 d of continuous inhibition which implies that sustained inhibition is needed for long-term cancer cell control [5]. Despite multiple efforts from researchers all around the world to discover potent inhibitors of LDHA, few viable inhibitors [1,6,7,8,9,10] emerged except pioneering work of GSK and Genentech. Their cellular effects or in vivo activity did not meet the clinical demands in spite of the potent biochemical activity [7].

Here, we report that reduction of LDHA by a synthesized organic arsenical causes bioenergetic and oxidative stress leading to cell apoptosis. The synthesized organic arsenical (PDT-BIPA) carrying the S-As-S displayed the best inhibition rates (*IC*_50/24 h_ = 0.55 ± 0.01 µM) for HL-60 cells among six cancer cell lines. PDT-BIPA can inhibit LDHA in HL-60 cells in a concentration- and time-dependent way leading to the burst of ROS and metabolic changes to suppress cancer cells and tumor in vitro and in vivo, respectively. Moreover, this compound manifested different inhibitory mechanism toward leukemia with our previous work [11,12].

## 2. Results

### 2.1. Synthesis

The medicinal power of arsenic is supported by strong evidence demonstrating As_2_O_3_ and some other organic arsenicals as successful therapeutic agents against APL (acute promyeloid leukemia [13]. Therefore, we aimed to develop more effective arsenic-based anti-cancer compounds. Inspired by the previous work [11,14], vanilline (which has no obvious medical effect at low concentration and short exposure times) was conjugated to a well-studied organic arsenical, PDT-PAO, to fabricate the target compound PDT-BIPA. This synthesis (shown in Scheme 1 and Appendix A) was very simple and caused little waste and pollution. The target compound demonstrated better efficacy and a new target protein inhibitory ability compared to PDT-PAO.

### 2.2. Cytotoxicity Screening

In order to assess the biological activity, the half-inhibitory concentrations (*IC*_50_) of PDT-PAO and PDT-BIPA for three leukemia cell lines (K562, NB4, HL-60), three other cancer cell lines (U87, HeLa, MCF-7), and one non-cancerous cell line (GES-1) were determined by MTT assay. As illustrated in Table 1, it could effectively inhibit the cancer cellular proliferation in a concentration-dependent way, and showed the best efficacy toward leukemia HL-60 cells (*IC*_50/24 h_ = 0.55 ± 0.01 µM). Furthermore, it displayed less toxicity toward non-cancerous cells (GES-1), indicating good cancer cell inhibitory capacity, as well as cellular selectivity. As a result, HL-60 cells were chosen for follow-up research. There was no significant difference in the levels of inhibition between PDT-PAO and PDT-BIPA and the *IC*_50/24 h_ value of vanilline toward HL-60 cells was over 100 µM.

### 2.3. LDH Inhibition

Lactate dehydrogenase (LDH), controlling the conversion of pyruvate to lactate, plays a key role in glycolysis, cell growth, and tumor maintenance. If LDH is inhibited, the metabolic pathway of the cancer cell will be disturbed to disorder. Indeed, the LDH activity was inhibited in a concentration and time-dependent way under PDT-BIPA treatment (Figure 1A,B). And the concentration of lactic acid (LD), the product of fermentation, also showed a decreasing tendency (Figure 1C,D). Additionally, the LDH and LD could be released to the surrounding extracellular environment. Therefore, the LDH and LD level of the medium were also tested. There was the same tendency of LDH and LD in the medium as within the cells (Figure 1E,F). The expression of LDHA, conversing pyruvate to lactate, decreased greatly (Figure 1G) while in contrast that of the LDHB clearly increased to accommodate the metabolic disturbances. In addition, the LDH and LD levels of normal cells GES-1 showed less inhibition even under high dosage exposure of PDT-BIPA (Appendix A) which demonstrated the highly selectivity of PDT-BIPA providing a good therapeutic window.

### 2.4. PDHC Alteration

PDHC (pyruvate dehydrogenase complex) catalyzes pyruvate to acetyl-CoA for tricarboxylic acid cycle whose activity is reduced by the increased cancer cellular PDK (pyruvate dehydrogenase kinase) expression. After the attenuation of LDH activity, the PDHC function was upregulated slightly, including the activity (Figure 2A,B) and the expression (Figure 2C), to accommodate the high speed extracellular nutrient uptake for proliferation. As the necessary disulfhydryl coenzyme, LA is associated with the function of dihydrolipoyl transacetylase (PDH E2), and is also a powerful antioxidant. That is, the LA (lipoic acid) may regulate the activity of PDHC. As shown in the figure, pretreatment of LA could alleviate the expression upregulation of PDHC and PDH E2 (Figure 2D) caused by PDT-BIPA incubation.

### 2.5. Alteration on Mitochondrial Respiration

The metabolism conversion could be evidenced by the alteration in enzyme function alteration and down-stream cascade reactions, such as the oxygen consumption rate (OCR) and the energy producing capacity. When intact, all cells showed almost the same oxygen consumption rate (Figure 2E), followed by a clear and instantaneous decrease in rate following PDT-BIPA treatment. After permeabilization by digitonin, and in the presence of sufficient main respiratory substrate-NADH, and secondary respiratory substrate-succinate, the O_2_ consumption rate showed a considerable increase. Cells pretreated with PDT-BIPA for 24 h displayed the same O_2_ consumption profile (Figure 2F). That is, in order to survive, the cancer cells up-regulated the OXPHOS to maintain the high speed of proliferation after fermentation was inhibited by PDT-BIPA. However, the inhibition by PDT-BIPA was strong enough to overcome this regulation and cause the low levels of O_2_ consumption in the intact cells. The ideal isolated mitochondria have integral mitochondrial inner membrane evidenced by the relatively low respiratory rate for state 4 and normal function of respiratory chain evidenced by the high respiratory rate for state 3 (about three-fold than state 4) (Appendix A). The exposure of PDT-BIPA brought about increases in both state 3 and state 4, however, resulting in a significant decline in respiratory control ratio (RCR = state 3/ state 4). These results manifested the comprehensive inhibition of PDT-BIPA toward mitochondrial respiration instead of the main alteration of LDH activity.

### 2.6. ROS Bursting and Glutathione and Thioredoxin Systems Regulation

It has been estimated that about 2% of the O_2_ that is consumed by the mitochondria is involved in ROS generation [15]. Attenuation of LDH and enhancement of PDH resulted in OXPHOS, which increased electron leakage from complex I and complex III leading to a corresponding increase in cellular ROS (Figure 3B) even after a short time and low dosage exposure of PDT-BIPA (Figure 3A). As a potential antioxidant, GSH was up-regulated to maintain ROS levels (Figure 3C left). When cells were incubated with its synthesis inhibitor BSO, the GSH level dropped sharply (Figure 3C right) which promoted the further bursts of ROS. In addition, GSH, coordinating metal ions with its thiol group, is involved in the removal of arsenicals through MRP1. As a result, the arsenical had low bioavailability in cancer cells and the MRP1 inhibitor MK571could increase the cellular arsenic concentration (Figure 3D). These two pieces of evidences demonstrated that increasing the levels of GSH could protect cells from the damage by PDT-BIPA (Figure 3E).

The Trx system helps by maintaining the redox balance and is also one of the targets of organic arsenicals. Here, PDT-BIPA exhibited the ability to inhibit TrxR activity at high dosage and long exposure time (Figure 3F). Besides, TrxR displayed down-regulated expression (Figure 3G). Combining the LDH inhibitory results, it could be proved that PDT-BIPA acted differently when compared with other arsenicals described in previous studies [12].

### 2.7. Impairment on Mitochondrion

High ROS exposure mainly from the electron transport chain could cause mitochondrial dysfunction, including the collapse of mitochondrial membrane potential, ATP synthesis obstruction, and membrane non-integrity. The burst of ROS induced marked a drop in mitochondrial membrane potential (Figure 4A), contributing to the reduction in ATP synthesis, which ultimately resulted in a lower ATP level (Figure 4B). Moreover, the isolated mitochondria reflected the same consequences: lower ATP levels (Figure 4C) and suppression of the heat output of isolated mitochondria (Figure 4D) reflected as the reduction in rate constant (*k*_2_ and *k*_3_) and reduced maximum power (Appendix A). Furthermore, the declined electron density of mitochondria was presented intuitively in the TEM pictures in cells after a long time and high dosage exposure (Figure 4E), and the irregular shape and occasional ruptured membrane of isolated mitochondria following incubation with PDT-BIPA for 40 min (Figure 4F). Taking all of these factors into consideration, the mitochondrial impairment was the result of LDH inhibition and ROS burst.

### 2.8. Induction of Apoptosis

On account of the alteration in mitochondrial respiration and the burst in ROS, cells can hardly retain their viability. Cytochrome *c* was released from the intermembrane space to initiate caspase activation in the cytosol. The content of cytochrome *c* ascended greatly in cytosol after the treatment of PDT-BIPA for 24 h in a dose-dependent manner (Figure 5A). At the same time, the expression of oncogenes such as C-myc and HIF-1α reduced to adapt the metabolic conversion (Figure 5D). In total, all these dysfunctions trigger apoptosis of the HL-60 cells. The apoptosis initiated from mitochondria evidenced by the increase of Bax and decease of Bcl-2 expression, followed by the activation of caspase 9, caspase 3, and the DNA repairing enzyme Parp. As shown in Figure 5B,C, the apoptosis proportion is expressed by the summation of FITC^+^/PI^−^ and FITC^+^/PI^+^. Further, pretreatment of cells with ZVAD-fmk, an inhibitor of caspase-mediated cascade apoptosis, blocked cell death to some extent while treatment with NEC-1, the inhibitor of necrosis, could not alleviate cell death.

### 2.9. Tumor Inhibition In Vivo

To examine the impact of PDT-BIPA on in vivo tumor growth, xenograft studies were performed using nude mice. After the mice born near 100 mm^3^ tumor, PDT-BIPA was given every two days in a dosage of 0.8 mg/kg or 1.6 mg/kg for the treated group four times (Figure 6A). As shown in Figure 6B,C, the tumor was dramatically reduced by PDT-BIPA in the 1.6 mg/kg group, as demonstrated by the volume (Figure 6E) and weight (Figure 6D) of tumor and the tumor inhibition ratio (Figure 6F) which got over 60%. On account of the malignant growth of the tumor, the body weight of the mice increased abnormally. However, PDT-BIPA could maintain the body weight at the normal level (Figure 6H). After the mice were sacrificed, some of the organs, tumor, and the femur were collected for the following assays. Together with the mice normal routine activity and the same spleen HE staining (Figure 7C) results of the three groups, the organ coefficient (Figure 6G) showed almost no side effects of PDT-BIPA treatment toward mice. The arsenic concentration in the femur (Figure 6I) was slightly increased in the 1.6 mg/kg group, as determined by ICP-MS, demonstrating the ability to control the leukemia cells in bone marrow which may lead to poor prognosis. Western blot analysis of the three groups demonstrated the same results with that in vitro (Figure 5E). Additionally, PDT-BIPA did not lead to tumor autophagy, as shown by the constant expression of LC3 and P62, key proteins involved in the autophagy progress. As shown in the figure, PDT-BIPA diminished the proliferation and augmented the apoptosis of tumor cells, a conclusion drawn from the increasing expression of Ki67 (Figure 7B: the count of stained cell with bright fluorescent signal is 16 in control group, 13 in 0.8 mg/kg group and 7 in 1.6 mg/kg group), the enhancement of TUNNEL signal (Figure 7D: the count of stained cell with bright fluorescent signal is 6 in control group, 15 in 0.8 mg/kg group and 15 in 1.6 mg/kg group), and the obvious appearance of apoptotic cells in the 1.6 mg/kg group of the tumor HE staining (Figure 7A).

## 3. Discussion

Cancer cells prefer glycolysis for some glycolytic enzymes showing that anti-apoptotic property and lactic acid promotes angiogenesis and facilitate metastasis to sustain the high speed proliferation. These altered pathways represent attractive therapeutic targets. The Warburg effect has been directly linked to the activation of oncogenes, such as *MYC*, *Ras*, and *Akt*, and loss of tumor suppressor genes such as *p53*, which result in the deregulated metabolic pathways and are responsible for the initiation of tumorigenesis and tumor progression [16]. The amplified myc family of genes in tumors including C-myc, L-myc, N-myc, and S-myc especially C-myc intensify the effects of growth-factor signaling on glucose metabolism and also regulate specialized metabolic activities for genome duplication [17]. Overexpressed myc led to the concurrent oxidation of glutamine via the TCA cycle and conversion of glucose to lactate [18]. In lymphomas, oncogenic myc activation causes a series of metabolism changes, like enhanced mitochondrial biogenesis, acceleration of mitochondrial glutaminolysis, and up-regulation of LDHA expression to drive glycolytic metabolism. HIF-1 targeting LDHA, glycolytic enzymes, glucose transporters, etc., requires the subunit HIF-1a to maintain activity, which is expressed under the control of PI3K/Akt/mTOR pathway [2]. HIF-1α facilitates cellular adaptation to an acid load by inducing expression of carbonic anhydrases. And the acidic pH enhances the activity of LDHA and MDH to a lesser extent which in turn consolidates the cellular adaptation [19]. PDT-BIPA regulated these two gene expression to rewire the metabolic pathway from glycolysis to OXPHOS.

Mounting evidences show that the tumor microenvironment regulates the balance between the immune response and tolerance by activating infiltrating cells, for instance, melanomas tumor surveillance by T and NK cells was inhibited by accumulated lactic acid resulting in the inhibition of differentiation and activation of monocytes and T cells in vitro. Tumor-derived lactic acid was not only for the tumor growth itself, but also for the immune cell balance in the tumor environment [20]. Studies indicated LDHA is responsible for tumor initiation evidenced by reduction of LDHA reduced cellular transformation and markedly delayed tumor formation. Attenuation or disruption of LDHA resulted in reduced tumor growth, and the authors attributed the diminished tumorigenicity to the compromised ability of tumor cells to grow under hypoxia or the importance of LDHA for tumor-initiating cells, respectively [21]. Tumorigenicity could be suppressed by LDHA knockdown accompanied with increased mitochondrial respiration [22]. LDHA inhibition results in increased apoptosis via ROS production in cell with fumarate hydratase deficiency and was viewed as a therapeutic strategy for the treatment of hereditary leiomyomatosis and renal cell cancer [23]. PDT-BIPA showed great inhibition to LDHA and in order to accommodate the change, the expression of LDHB was increased ending up this metabolic alteration.

The increase in mitochondrial respiration is associated with an increase in production of mitochondrial reactive oxygen species and normalizing mito-membrane potential, which would promote mitochondria-dependent apoptosis.

Toxic levels of ROS production contribute to oxidative stress and cell death. Thus, one of the potentially effective cancer therapies may be elevating ROS production [24]. Chemotherapy, such as anthracyclines, cisplatin, arsenic trioxide is widely used in cancer by increasing ROS production resulting in irreparable damage and cell death [25]. Redox control systems, including enzyme and nonenzyme systems, for instance, catalase, glutathione peroxidase (GPX), peroxiredoxins (PRX), thioredoxin (Trx), GSH, and NADH, maintain ROS levels [26]. Here, GSH was increased to scavenge the abnormal ROS level. And also Trx system as one of the targets of arsenicals was regulated under PDT-BIPA exposure.

Mitochondrial hyperpolarization implied an apoptosis resistance state for the efflux of pro-apoptotic mediators through the mitochondrial transition pore depends in part on mitochondrial membrane potential. Thus, the PDH-BIPA induced the mitochondrial depolarization might get rid of the cancer cellular ability of inhibition of mitochondria-dependent apoptosis.

## 4. Materials and Methods

### 4.1. Chemicals

*p*-Arsanilic acid, 70% ammonium thioglycolate solution, and 1,3-propanedithiol, *L*-buthionine-(S,R)-sulfoximine (BSO) and β-nicotinamide adenine dinucleotide (NADH) were purchased from Aladdin (Shanghai, China). RPMI 1640 Medium, Dulbecco’s modified Eagle Medium (DMEM) and fetal bovine serum (FBS) were obtained from GIBCO (Grand Island, NY, USA). (±)-α-lipoic acid (LA), Hoechst 33342, propidium iodide (PI) and 2′,7′-dichlorfluorescein diacetate (DCFH-DA) were obtained from Sigma-Aldrich (St. Louis, MI, USA). 3-(4, 5-dimethylthiazol-2-yl)-2,5-diphenyltetrazolium bromide (MTT) was obtained from Amresco (Solon, OH, USA). MK571 was obtained from Selleck (Shanghai, China). Mitochondria Staining Kit (JC-1) and Annexin V-FITC/PI apoptosis kit were purchased from MultiSciences (Hangzhou, China). Tetramethylrhodamine (TMRM) was obtained from HEOWNS (Tianjin, China). Human Cyt *C* ELISA Kit and Human PDH E2 ELISA Kit were obtained from Elbio (Shanghai, China). RIPA buffer, BCA Protein Assay Kit, ATP Assay Kit, and dihydroethidium (DHE) were purchased from Beyotime (Shanghai, China). Human PDHC Assay Kit was purchased from Letter Sail (Nanjing, China). A 10-mM solution of PDT-PAO-F16 was prepared in DMSO and stored at 4 °C. Other common chemicals were of analytical reagent grade from Shenshi (Wuhan, China) and used without further purification.

### 4.2. Synthesis and Characterization of 4-(1,3,2-dithiarsinan-2-yl)aniline (PAO-PDT)

*p*-Arsanilic acid (4 g, 18 mM) was stirred in 70% ammonium thioglycolate (10 mL, 13 g, 120 mM) at 50 °C. After 2 h, 1,3-propanedithiol (4 mL, 23 mM) was added drop wise. After 4 h, the turbid liquid was extracted with DCM about 10 times. The solution of DCM was condensed for the silica gel column chromatography (PE/DCM = 1/1, v/v) to give the product 52% yield. ^1^H NMR (400 MHz, DMSO-d6) δ: 7.44–7.42 (d, J = 8.0 Hz, 2H), 6.69–6.67 (d, J = 8.0 Hz, 2H), 5.55 (s, NH_2_, 2H), 2.88–2.78 (m, 4H), 2.05–1.95 (m, 1H), 1.90–1.80 (m, 1H). ^13^C NMR (400 MHz, DMSO-d6): δ: 150.88, 133.78, 121.32, 114.82, 28.91, 27.11. ESI-MS: calcd for C9H12NS2As: 272.9627, found 273.9708 (M + H)^+^.

### 4.3. Synthesis and Characterization of 2-methoxyl-4-(((4-(1, 3, 2-dithiarsinan-2-yl) phenyl) imino) methyl) phenol (PDT-BIPA)

PAO-PDT (5 mM) and vanilline (6 mM) were dissolved in 20 mL of anhydrous ethanol and stirred for 2–4 h under reflux. The resulting mixture was cooled gradually to yield the yellow crystal. Then the crystal was washed with DCM and recrystallized in ethanol to give the product in 61% yield. ^1^H NMR (400 MHz, DMSO): δ 8.50 (s, 1H), δ 7.84 – 7.82 (d, J = 8 Hz, 2H), δ 7.55 (s,1H), δ 7.38 – 7.36 (m, 3H), δ 6.92 – 6.90 (d, J = 8 Hz, 1H), δ 3.85 (t, 3H), δ 2.79 (m, 4H), δ 1.98 – 1.93 (m, 2H). ^13^C NMR (400 MHz, DMSO-d6): δ: 161.69, 153.37, 150.99, 148.48, 134.52, 133.63, 128.15, 124.98, 122.34, 115.82, 110.79, 55.99, 28.34, 26.18. ESI-MS: calcd for C_17_H_18_AsNO_2_S_2_:406.9995, found 408.0070 (M + H)^+^.

### 4.4. Cell Culture

HeLa, U87, MCF-7, and GES-1 cells were cultured in DMEM supplemented with 10% FBS and 1% penicillin/streptomycin solution in an atmosphere of 5% CO_2_ at 37 °C. NB4, HL-60, and K562 cells were cultured in RPMI 1640 Medium under the same conditions.

### 4.5. MTT Assay and Trypan Blue Assay

Eighty microliter (1 × 10^5^ /mL) cells were cultured in 96-well plates overnight followed by adding 80 μL related medium containing compounds of different concentrations for 24 h or 48 h. The percentage of DMSO in all conditions was under 0.1% (V/V) and cells incubated with DMSO only were used as controls. In the end, 40 μL MTT (2.5 mg/mL) was added to each well and incubated for another 4 h in the incubator. For the NB4, HL-60, and K562 cells, 100 μL 10% SDS solution was added, and the plates were placed in the dark at room temperature. 12 h later, the absorbance was measured at 490 nm and 570 nm using a microplate reader. As for the other adherent cells, 150 μL DMSO was added into each well for at least 15 min in the dark at 37 °C to dissolve the cells after the medium was removed for the following absorbance detection (BioTek, Winooski, VT, USA).

To investigate the GSH protection, GSH of different concentrations was added to the cells when they were placed into the 6-well plates for overnight incubation. Then the PDT-BIPA was added to the medium for 24 h treatment followed by being collected and being washed once with PBS. Next, the cells were stained with trypan blue (0.4%, *w*/*v*) and the number of live (non-stained) and dead (stained) cells were counted under the microscope.

### 4.6. Determination of LDH Activity and the LD Concentration

The cell was collected by centrifugation (1500 rpm, 4min) and all the related medium samples were collected in another group of tubes. For cells, the cells were washed with PBS once before lysed following the kit instruction. Then the lysate or the related medium was tested as the samples. For the LD concentration test, the same sample with the LDH activity assay was treated with protein precipitant as the sample pretreatment. All the tests were carried out following the kit instruction (Cominbio, Suzhou, China).

### 4.7. Measurement of PDHC Activity

A total of 75 μM LA was added to the medium with cells before the overnight incubation. After HL-60 cells were incubated with different concentrations of PDT-BIPA for different time, the cells were collected and washed once with PBS. Then the operation was conducted based on the kit instruction (Cominbio, Suzhou, China), except the time point setup, which was 10 s every point instead of 3 min.

### 4.8. Respiration

The oxygen consumption rate (OCR) was measured with Clark Oxygen Electrode (Hansatech Instruments, Norfolk, UK).

As for the cellular OCR, the cells were incubated with PDT-BIPA for 24 h before harvesting to get 1 × 10^6^ mL^−1^ in 1 mL cellular buffer solution (250 mM sucrose, 2 mM K_2_HPO_4_, 5 mM MgCl_2_, 1 mM EDTA, 20 mM Mops, and 1 mM ADP) for the determination of the influence of PDT-BIPA. Then the cell was permeated by 0.1 mg/mL digitonin to dissipate the respiratory substrates, all the cells came to almost respiratory arrest. Next, the respiration was initiated with 5 mM NADH substrate to complex I followed by being provided 5 mM succinate substrate to complex II. Each state was recorded for about 2 min. All operations above were performed under 37 °C and magnetic stirring.

As for the isolated mitochondria, mitochondria (1 mg protein) extracted based on the literature was added into 1 mL mitochondrial buffer solution (100 mM sucrose, 10 mM Tris, 10 mM Mops, 2 mM MgCl_2_, 50 mM KCl, 10 mM K_2_HPO_4_, 1 mM EDTA, 5 mM succinate, and 2 μM rotenone). State 4 was initiated by the addition of 1 M succinate only and state 3 was initiated by adding 1 M succinate and 250 μM ADP into the mitochondrial buffer solution. All operations above were performed under 25 °C and magnetic stirring.

Female Wistar rats weighing 130–150 g purchased from the Hubei Center for Disease Control and Prevention (Wuhan, China) were kept in micro-isolator cages with free access to water and food in a temperature-controlled room (22 ± 2 °C). The isolation of rat liver mitochondria was performed in compliance with the Guidelines of the China Animal Welfare Legislation, which has been approved by the Committee on Ethics in the Care and Use of Laboratory Animals of the College of Life Sciences, Wuhan University. (approved date: Oct., 22nd, 2019)

### 4.9. Measurement of TrxR Activity

HL-60 cells were incubated with 2 μM PDT-BIPA before being collected and lysed for the TrxR activity assay. And also the time point was set as 10 s every point and the other operations were followed the kit instruction.

### 4.10. Measurement of the Intracellular ROS Level

HL-60 cells treated with PDT-BIPA of 0.25, 0.5, 1, and 2 µM were seeded in 6-well plates for 24 h or cells with 0.5 µM PDT-BIPA were incubated for 4, 8, 12, and 24 h. As for the BSO pretreatment groups, 50 µM BSO was added to the cells when they were placed in the plate for overnight. Then the cells were collected and washed once with PBS followed by being incubated with PBS containing DCFH-DA (0.2 µM) for 30 min in dark at 37 °C. Next the cells were collected and washed once with PBS. Data were obtained and analyzed using a C6 flow cytometer (BD Biosciences, Franklin Lakes, NJ, USA).

### 4.11. Assessment of the GSH Level

Cells with or without 50 µM BSO were seeded in 6-well plates overnight and then treated with 0.5 μM PDT-BIPA for different time. Then the cells were collected and washed once with PBS before being lysed. The protein content was quantified by BCA kit (beyotime, Shanghai, China). All the operations were conducted according to the manufacturer’s instructions (beyotime, Shanghai, China). The absorbance was measured every minute at 405 nm with microplate reader (BioTek, Winooski, VT, USA).

### 4.12. As Concentration in Cells

Cells with or without 50 µM MK571 were incubated in 10 cm dishes overnight followed by 10 µM PDT-BIPA treatment for 6 h. Then all the cells were collected and washed twice with PBS. Total of 1 mL concentrated nitric acid was added to the cells and the solution was transferred to glass tube before being dried at 150 °C to reduce all the organic matters. Next, the solid was dissolved by 2 mL (0.1 mM) nitric acid and the solution was filtered by 0.22 µm membrane filter for the ICP-MS assay.

### 4.13. Assessment of Mitochondrial Membrane Potential

HL-60 cells treated with PDT-BIPA of 0.25, 0.5, 1, and 2 µM were plated in 6-well plates for 24 h or CCCP for 1 h. Then the cells were collected and washed once with PBS followed by being incubated with PBS containing TMRM (1 µM) for 30 min in dark at 37 °C. Next the cells were collected and washed once with PBS. Data were obtained and analyzed using a C6 flow cytometer (BD Biosciences, Franklin Lakes, NJ, USA).

For the isolated mitochondria, the change of mitochondrial membrane potential was monitored by observing the change in fluorescence emission intensity of 0.25 mM Rh123.

### 4.14. Measurement of Intracellular ATP Level

After HL-60 cells were incubated with PDT-PAO-F16 (0.25, 0.5, 1 and 2 µM) for 24 h, the cells were collected for the following tests according to the manufacturer’s instructions. Data were obtained and analyzed on a Multimode Plate Reader VICTORTM X5 (PE, Melville, NY, USA).

### 4.15. TEM

As for cell TEM tests, cells were incubated with or without PDT-BIPA for 24 h and then collected without any more wash. Next, 1 mL 2.5% glutaric dialdehyde solution was added to the cells for the following sample preparation and test.

As for isolated mitochondria TEM tests, mitochondria were suspended in Solution B’ (0.22 M mannitol, 0.07 M sucrose, 1 mM EDTA, pH 7.4) and incubated with PDT-BIPA for certain time before the collection by centrifuge and re-suspended in Solution B’ for further testing.

### 4.16. Micro-Calorimetry

The metabolic thermogenic curves of isolated mitochondria were obtained with ampoule method in the Thermal Activity Monitor III (TAM III, TA Instruments, New Castle, DE, USA). A total of 1 mL buffer in an ampoule containing Solution B’, 5 mg protein mL^−1^ mitochondria, 15 mM pyruvate, and various concentrations of PDT-BIPA prepared under 4 °C were put into the TAM III instrument at 30 °C. The data collection was terminated after the thermogenic curve came to the plateau.

### 4.17. Determination of Cellular Cytochrome c Level

HL-60 cells incubated with PDT-BIPA (0.25, 0.5, 1 or 2 µM) for 24 h were collected and washed once with PBS. Then the cells were sacrificed and dealt with ELISA Kit according to the protocol of the manufacturer.

### 4.18. Apoptosis

HL-60 cells were seeded in 6-well plates with or without 20 µM ZVAD-fmk or 0.49 µM NEC-1 overnight and treated with PDT-BIPA for another 24 h, then the cells were collected and washed by PBS once for the following tests. The incubation time of the two probes was 30 time instead of 5 min or other.

### 4.19. Western Blotting

Total cell protein extracts were prepared with RIPA containing PMSF at a ratio of 1000:1. Protein concentration was measured by BCA Protein Assay Kit (Beyotime, Shanghai, China). In total, 20–40 μg protein were separated by 8–12% SDS-polyacrylamide gel electrophoresis, transferred onto the polyvinylidene difluoride membrane, and blocked with 5 % fat-free milk for 2 h. The primary antibodies were incubated with membranes at 4 °C overnight, followed by HRP-conjugated secondary antibodies for 1 h at room temperature. Then the images were snapped by QuickChemi5100.

### 4.20. Murine Model Establishment and Drug Treatments

Animal experiments were conducted according to the guidelines of the Laboratory Animal Center of the Wuhan University College of Chemistry and Molecular Sciences (WH2019H008, Oct., 22nd, 2019). All experiments were conducted under approved procedures.

The mice (BALB/c, nu/nu, female, 5 weeks) purchased from Beijing Vital River Laboratory Animal Technology Co., Ltd. were injected with 0.1 mL HL-60 cells (2.0 × 10^7^ /mL) in the right oxter. After the tumor reached approximately 100 mm^3^, the mice were randomly divided into three groups: control (cyclodextrin: DMSO = 8:2) and 0.8, or 1.6 mg/kg every two days (i.p.) for four times. Tumor growth and the body weight of the mice were monitored every two days.

### 4.21. Tumor Hematoxylin and Eosin (H&E), Immunohistochemistry (IHC), and Western Blot

The tumors and spleens were formalin-fixed for 24 h and decalcified for 3.5 h, paraffin-embedded, cut, dewaxed, hydrated, some samples were stained with H&E, other samples were under antigen retrieval, then blocked with primary antibody, the next fluorophore-conjugated secondary antibody. Part of the tumors were collected and lysed by RIPA containing PMSF at a ratio of 1000:1. Protein concentration was measured by BCA Protein Assay Kit (Beyoutime, Shanghai, China). And the next operation was the same as the Western blotting of cells.

### 4.22. Statistics

All experiments were performed at least for two replicates. The data were presented as mean ± SD. The independent Student’s *t* test was used to compare the means of two independent groups. Significance was set at *p*** < 0.05, *p**** < 0.01 and *p***** < 0.001.

## 5. Conclusions

The synthetic organic arsenical PDT-BIPA-manifested inhibition toward cancer cells and tumor in xenograft model by remodeling metabolic pathways leading to getting rid of cancer cell advantages and to apoptosis. Besides, it merely demonstrated any inhibition toward enzymes except LDH which the normal cells depend scarcely on, and less toxicity toward normal cells (over 10 folds less than HL-60 cells). There is a therapeutic window between proliferating cancer cells and proliferating normal cells. In other words, PDT-BIPA is a potential and powerful candidate for the development of successful cancer therapies targeting metabolic pathway.

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
