# Peer review of "LDHA Suppression Altering Metabolism Inhibits Tumor Progress by an Organic Arsenical"

_ijms, 2019, doi:10.3390/ijms20246239_

Round 1
Reviewer 1 Report
The authors identify a new organic arsenical for utilization as an anti-cancer therapy. The study is well-executed and shows efficacy in a nude mouse xenograft model of tumorigenesis. I would like to see a sentence discussing that the decrease in body weight in the 1.6 mg/kg treatment could also be due to toxicity. In the experimental paradigm alone, it cannot be excluded. There are also some moderate English grammar changes that should also be examined. The study is appropriate and well-suited for IJMS.
Author Response
Hi reviewer,
Thank you for your kind help with our work.
For the first reminding about the body weight of the 1.6 mg/kg group, after carefully check of the original data and the body weight curve, a small mistake was made by calculation of the last point of the body weight of this group. We apologize for our mistake and thank you for you careful work. Here is the original data of the body weight and I will correct the curve in figure 6H. As mentioned in the main text, together with the mice normal routine activity, the spleen HE staining and the organ coefficient, these results manifested almost no side effect of PDT-BIP towards mice.
|
Mice number |
1 |
2 |
3 |
4 |
5 |
6 |
average±SD |
|
Weight/g |
16.10 |
16.29 |
15.46 |
16.06 |
21.59 |
17.67 |
17.20±2.27 |
For the second reminding about the English grammar, we will check the main text carefully and ask English native speaker for help to improve our paper.
Many thanks.
Reviewer 2 Report
This manuscript describes the evaluation of a synthetic organoarsenic compound as an anti-cancer agent. The authors synthesized PDT-BIPA from PDT-PAO which is a well-studied organoarsenic compound. PDT-BIPA indicated slightly higher cytotoxicity and LDH inhibitory activity to cancer cell lines than normal cell line (GES-1). In Hl-60 cells, the treatment of PDT-BIPA indicated a decrease in the protein level of HDLA. PDT-BIPA indicated several effects such as the upregulation of the expression level of PDHC and PDH E2, and the increase of OCR. The increase of mitochondrial respiration caused ROS bursting, followed by a disorder of mitochondria, and apoptosis. Furthermore, PDT-BIPA indicated a decrease of the tumor in xenograft studies with mice. Since PDT-BIPA has potential as an anti-cancer agent, this manuscript is recommended for publishing after minor revision.
Minor points:
Figure 4, F: Could you add the explain the differences in cell image of above and below in this article. Figure 7, B and D: Since it is difficult to decide the increasing expression of Ki67 and the enhancement of TUNNEL signal by picture, could you challenge to quantify fluorescence intensity.Author Response
Hi reviewer,
We appreciate very much of your time for reviewing our manuscript and of your valuable suggestions and comments.
For the first suggestion about the figure 4F, the above and the below two pictures are the same group. Sorry for the unclear statement and we will add this statement to the manuscript.
For the second suggestion about the Ki67 and the TUNNEL pictures, there may be no significant fluorescence intensity difference. But the stained cell number changes. Thanks for your kind suggestion, we will add this explain to the manuscript.
Cheers!